# Dynamic Properties of Soil Cements for Numerical Modelling of the Foundation's Basis Transformed under the Technology of Deep Soil Mixing: A Determination Method

Armen Ter-Martirosyan *[ID], Vitalii Sidorov and Evgeny Sobolev [ID]

Federal State Budget Educational Institution of Higher Education, "National Research Moscow State University of Civil Engineering" (NRU MGSU), Jaroslavskoe Shosse 26, 129337 Moscow, Russia; vitsid@mail.ru (V.S.); soboleves@mgsu.ru (E.S.)
* Correspondence: gic-mgsu@mail.ru

**Abstract:** This research investigates the mechanical properties of soil-cement specimens ranging from ultrasmall to large values of shear strain at dynamic loading. The nonlinear behavior of soil cement exposed to dynamic loading in a wide range of changing shear strains was examined on the basis of two mechanical models. All soil-cement specimens were collected from under an existing building and modified with deep soil mixing (DSM). Soil-cement samples were examined using low-amplitude oscillations in the resonant column and the dynamic triaxial compression method. Additionally, the stress–strain state for modified footings exposed to dynamic loading, and the approximation of soil stiffness and damping coefficient was analyzed. Dependencies on the basis of the resilient elastic models of Ramberg–Osgood and Hardin–Drnevich are proposed for application. Results reveal that the empirical graphs of the dependency soil stiffness–shear strain based on various methods exhibited the distinctive S-shape of decreased stiffness. The stiffness of the soil cement was reduced by 50% of the maximal value at shear strains of the $10^{-3}$ decimal order. The method presented in this study enables the drawing of stiffness change and damping–shear strain dependency where the range of shear strains changes from ultrasmall to large strains. The normalized modulus of shearing and the damping coefficient on shear strains for soil cement could be obtained under the proposed method. This method can be used for the preliminary calculations of structures on the footing modified by mathematical modelling or when field research data from site investigation are not available.

**Keywords:** stiffness; damping; soil cement; dynamic loading; dynamic triaxial compression test

## 1. Introduction

Geotechnical forecasting and building foundation analysis under dynamic loading are not conceivable without an accurate definition of the mechanical properties of soils. This statement is also applicable to a modified foundation's basis. Deep soil mixing technology has become widespread in many countries. This transformation emerges when the basis is produced with a specific material called soil cement. The stiffness of soil cement and its ability to absorb energy demonstrate a nonlinear dependency on shear strain when the ultrasmall strains range from $10^{-4}$ to $10^{-2}$%. The strain changing range under dynamic loading is significant for shear strains, and varies between $10^{-4}$ and $10^{-2}$%. It is necessary to investigate the material's behavior within the entire range of shear strain changes for a reliable forecast of a modified soil basis. The distinctive nature of complex research is necessary to combine the results of tests conducted with the methods of dynamic triaxial compression and low-amplitude oscillations in resonant columns to determine the dynamic behavior of soil cement. This task becomes more difficult due to the specific physical and mechanical properties of soil cements, such as the dependency of the obtained parameters on humidity and density, limitations on the applicability of various sensors for recording

displacements when exposed to overpressure, humidity, and the stringency of the operating chambers of triaxial compression apparatuses.

The behavior of geological conditions due to seismic hazards is essential to forecasting the impact of dynamic property changes. The stress–strain behavior (SSB) of transformed soils under foundation increases the responsibility of designing, constructing, and maintaining buildings and structures. It is required to factor in various types of dynamic loading of natural and technogenic character. The proposed research is of paramount importance for industrial structures (nuclear power plants and hazardous chemical factories); their irregular operation could lead to substantial public, ecological, and economic consequences. The research results enable new methods of studying and forecasting stress–strain behavior of the modified bases of such buildings and structures in geotechnical design. The experimental investigation of the dynamic properties of transformed soils allows for enhancing the reliability of design, construction, and maintenance for complex engineering–geological conditions, and reducing technical and economic expenses arising during design in construction.

One of the principal geotechnical tasks in designing buildings is to prevent the development of a soil base's joint deformations and a structure beyond the permissible limit values, standardized by Russian and international regulatory documents. As a rule, to reach the calculation values of strains and ensure the dynamic stability of soils without moving a structure to another construction site, measures must be undertaken to transform the soil base's properties. The well-known deep soil mixing method (DSM) is applied widely for the transformation of the construction properties of soil bases (Romanov and Racinais, 2018) [1]. This technology has an advantage over similar methods of jet grouting and pile bottom foundation work while applying a lesser amount of a binder (cement). Few international control documents regulate works execution; for instance, European Standard 14679:2005 recommends executing special geotechnical works and deep mixing. The behavior of soil bases modified under the DSM method against static and dynamic loading has not been studied extensively. Therefore, the issue of the interaction of a modified soil base and a structure exposed to static loading requires further study (Khosravi et al., 2016) [2].

This issue has lately been given much attention in various studies. The impact of soil base transformation with the DSM method on stiffness under seismic loading was studied by Shaghaghi et al. (2021) [3]. Results indicated that the soil base stiffness is increased upon its transformation; the high stiffness of the binding material preconditions this. The study mentions the high cost of this procedure due to the significant volume of soil mass to be transformed for primary structures. Nevertheless, this approach is still much cheaper than the conventional schemes applying the seismic isolation system. Tiwari and Upadhyaya (2017) [4] reported that the stress in the soil massif, when exposed to seismic load, is reduced by 48% due to 30% soil being replaced by soil cement. Researchers Namikawa and Usui (2019) [5] revealed that the stiffness of a transformed soil base increased, and stated that soil stiffness was increased by approximately 20% at impulsive loading, corresponding to the most intense seismic impact.

The investigation of various factors affecting soil cements' strength and dynamic properties was thoroughly studied by Kim et al. (2018) [6]. Findings indicated that the shearing action and soil-cement stiffness were increased with time of strength generation. Whereas the properties depend on stress (generic shear modulus and damping coefficient), different trends were observed where stiffness was regarded as the function of the time of strength generation and the content of a binder. The results of prior studies by Li et al. (2016) [7] demonstrated that dynamic shear strain develops at a lower rate while dynamic strength gradually rises with the increase in the fraction of a binder in the composition of a soil–cement mixture. The damping coefficient of soil cement was decreased as a function of growing cyclic stresses and the number of cycles under cycling loading.

A semilogarithmic model fully described the degradation of soil cement's shear modulus and damping coefficient. The importance of the relationship between the binder's content and strength gain time in geotechnical properties of modified soils is emphasized by Upadhyaya et al. (2016) [8]. Studies by the triaxial compression method implemented

by Du et al. (2021) [9] proved that the confining pressure was not significantly impacted the generic modulus of elasticity for soil cement; it was determined by the content of a binder and strength gain time. The same work noted that the damping coefficient is reduced with the increase in confining pressure for the specific content of a binder and strength gain time. The dependency of the damping coefficient obtained in this research for various binder contents, strength gain times, and confining pressures corresponded precisely to the exponential function. Ter-Martirosyan and Sobolev (2020) [10] studied the impact of changes in humidity, density, and confining pressure on determining the dynamic properties of soil cement. Laboratory tests by the triaxial compression method were implemented at the specified humidity of soil-cement specimens (air-dry specimens, natural water content, and water-saturated ones). The anisotropic stressed state of the soil-cement specimens exposed to triaxial tests in the resonant column was preconditioned by the specific features of the foundation of the designed building. The research showed that additional vertical loading is the essential factor affecting stiffness. Ter-Martirosyan and Sobolev emphasize that, with increasing density, the velocity of transverse waves decreases. Humidity is directly connected with soil-cement density and the amount of water in pores, so the evaluation of its impact on the dynamic properties of soil cement is analogous to the influence of density change. Sobolev and Ter-Martirosyan (2020) [11] proposed a method for controlling the stress–strain behavior of a transformed soil base exposed to dynamic loadings. Similar research was implemented by Wei et al. (2020) [12], who presented systematic laboratory testing for the impact of water content on soil-cement density while controlling the density and porosity coefficient. The influence of the porosity coefficient and a binder's content was investigated. The strength of sandy soil stabilized by cement was constantly lowered, while the water/cement ratio increased from 0.5 to 3. The general equation was proposed for the evaluation of soil-cement strength. The water/cement ratio of the soil stabilized by cement significantly impacts strength and stiffness, as reported by Wang et al. (2022) [13]. The investigation of the new method of water/cement ratio's selection for construction with the DSM method. The empirical model for evaluating the optimal water/cement ratio of the transformed soil with variable cement content was developed on the basis of empirical data.

Apart from purely experimental approaches to studying the inter-relation of the physical properties of soil cement, its stressed state and dynamic properties, there are approaches based on analytical analysis that study the interaction of discrete particles of soil cement in the shape of spheres (discrete mechanics). An example of this investigation is Wei et al. (2021, 2022) [14,15], who examined the advantage of semianalytical one-parameter correlations between the velocity of waves' propagation and the stressed state or correlation between the stiffness of soil cement and stressed state. Notwithstanding mathematical rigor, the analytical approach is to be verified by an experimental laboratory and field tests. The study of damping of the dynamic soil shear modulus for various types of soils, both natural and modified, is an urgent task in designing high-speed railways. Study Ye et al. (2021) [16] is performed by a generalization the empirical models of the dynamic modulus and the range of damping curves. A method for determining the value of the dynamic modulus with a reasonable range of working shear deformations is proposed.

Nguyen et al. (2021) [17] stated the importance of obtaining the parameters of cement deep mixing DSM columns for numerical finite element modeling, and comparing results with geotechnical monitoring. Viet et al. (2020) [18] considered two models that define the soil. The models of the Mohr–Coulomb (MC) and soft soil (SS) were used to model the behavior of deep excavation, the retaining wall, and base, which were improved with the use of deep cement mixing columns (CDMs). Results presented in this study can be used in similar studies for these models. Dang et al. (2018) [19] performed similar studies that also required determining damping parameters and a decrease in the stiffness of soil cement with increasing shear deformation. Rashid et al. (2015) [20], and Priol et al. (2007) [21] stated that, despite the long history of soil cementation technology, the need to determine the parameters under dynamic effects is especially relevant for practical use. In the studies

under consideration, numerical analysis based on the effective strength parameters was carried out, which gave a more realistic idea of the real strength of the reinforced base and its interaction with the surrounding soil. This simulation was perfomed with a 3D finite element method model to provide a better reinforcement estimate.

This study explores the mechanical characteristics of soil-cement specimens with shear strains ranging from ultrasmall to large. On the basis of the two mechanical models, the researchers did not fully explore the nonlinear behavior of soil cement subjected to dynamic loading in the broad range of changing shear stresses. The current investigation was performed to fulfill this research gap, which is considered to be the novelty of this research.

## 2. Materials and Methods

In laboratory conditions, soil modified by cement samples from the designed nuclear power plant site was extracted and tested using a triaxial dynamic test. Before the modification process, were presented fine sands of alluvial genesis. The reactor building was constructed in situ and the soil basis is constructed using DSM technology. As a binder, Portland cement V was used in compliance with ASTM C150/C150M-19a Standard Specification for Portland Cement (compression strength at 28 days is 21 MPa). The undisturbed samples of cement-stabilized soils whose height was 140 mm and diameter 70 mm were used, as shown in Figure 1. The principal physical properties of soil cements were determined during the investigations and are demonstrated in Table 1.

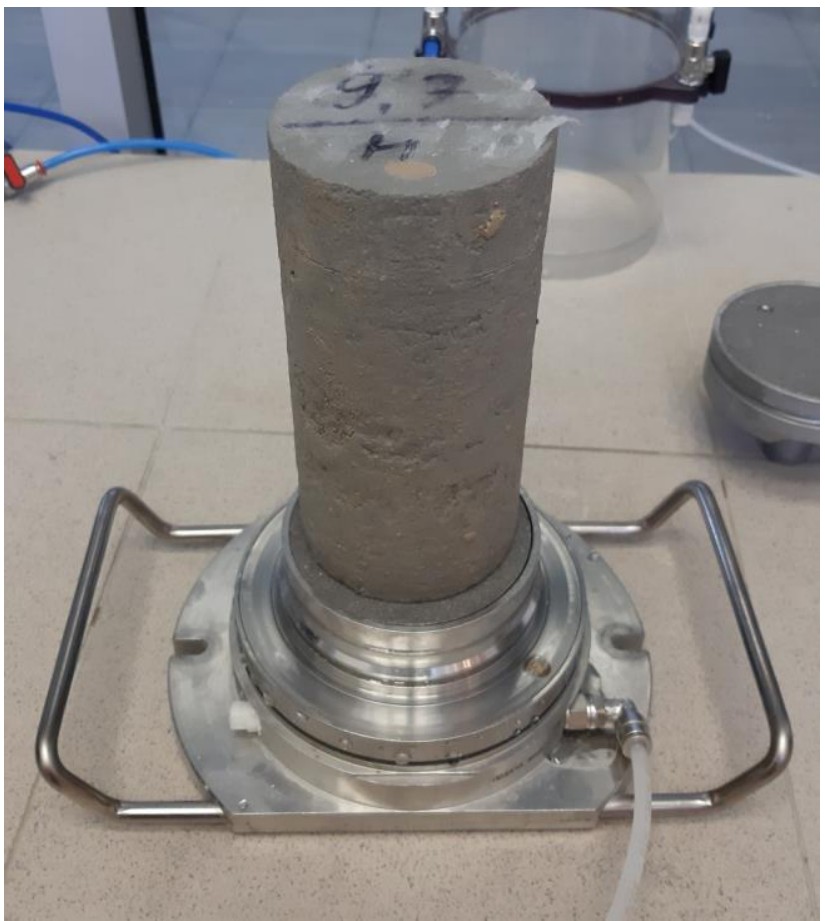

**Figure 1.** A fully water-saturated soil-cement specimen on the dismountable lower pedestal of the resonance column.

**Table 1.** Physical properties of the investigated cement-stabilized soils.

| S. No. | Depth of Specimen Collection H, (m) | | Humidity W, Unit Fraction, Containers | Density, g/cm³ | | | Porosity Coefficient, Unit Fraction | Moisture Level Sr, Unit Fraction |
|---|---|---|---|---|---|---|---|---|
| | From | To | | In a Natural State | Dry Soil | Soil Particles | | |
| 1 | 10.2 | 10.6 | 0.641 | 1.56 | 0.95 | 2.66 | 1.8 | 0.95 |
| 2 | 18.1 | 18.3 | 0.514 | 1.67 | 1.1 | 2.67 | 1.43 | 0.96 |
| 3 | 19.1 | 19.3 | 0.600 | 1.59 | 0.99 | 2.66 | 1.69 | 0.95 |
| 4 | 3.7 | 4.0 | 0.700 | 1.54 | 0.91 | 2.66 | 1.92 | 0.97 |
| 5 | 7.1 | 7.4 | 0.602 | 1.59 | 0.99 | 2.67 | 1.7 | 0.95 |
| 6 | 9.0 | 9.4 | 0.696 | 1.53 | 0.9 | 2.67 | 1.97 | 0.94 |

The modified soil samples were tested using the low-amplitude oscillations in the resonant column method [22], as shown in Figure 2a. Three dynamic triaxial compression tests (as shown in Figure 2b) were conducted (Seed and Idriss, 1970) [23] to study the nonlinear dynamic behavior of soils and plot the degradation curves (dependency of shear modulus $G$ and damping coefficient $D$ on shear strains $\gamma$ at cyclic loads).

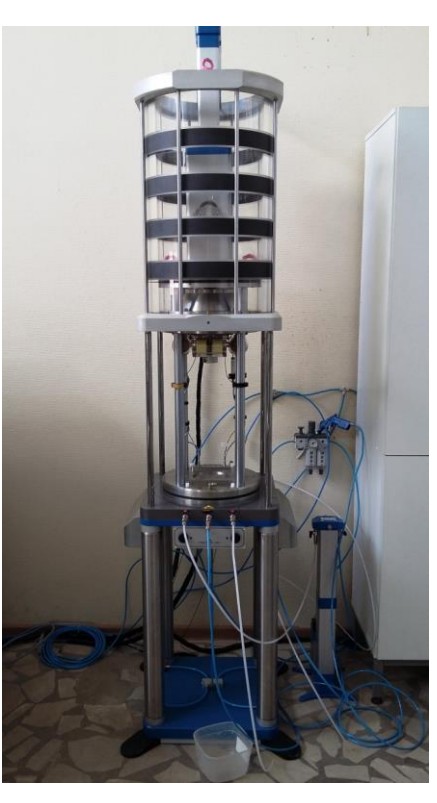
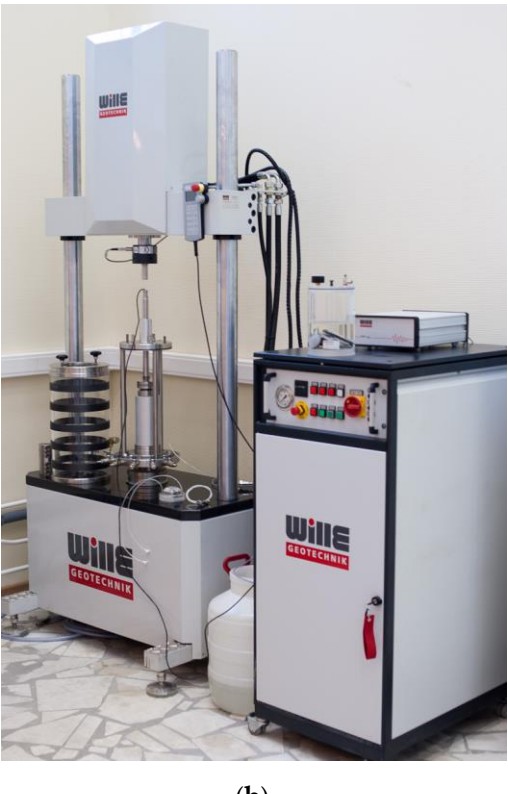

(**a**)        (**b**)

**Figure 2.** Laboratory equipment for testing dynamic properties of cement stabilized soil. (**a**) Resonant column and (**b**) the apparatus of dynamic triaxial compression.

The laboratory unit for dynamic triaxial compression was produced in Germany, Gottingen, APS Antriebs-, Prüf- und Steuertechnik GmbH, model LO70-SH0063-S2 (https://www.wille-geotechnik.com; accessed on 1 June 2022). The laboratory setup for testing using low-amplitude torsional vibrations (resonant column) was produced in Russia by NPP Geotek LLC, model GT 1.3.3. All used laboratory equipment was standard, and passed metrological control and verification. The implemented studies allowed for deriving the set of dependencies describing the decrease in $G$ and increase in $D$ on shear strain $\gamma$; these

dependencies enable foreseeing the stiffness change of soils exposed to dynamic loadings corresponding to the predicted seismic impact.

The actual shear modulus values obtained during the experiments were normalized to the value of initial shear modulus $G_0$. The value of shear modulus $G$, obtained on the basis of test results in the resonant column at low shear strains $\gamma$ from $10^{-4}$ to $10^{-3}$%, was assumed to be initial shear modulus $G_0$, i.e., under the first most significant value of shear modulus. All calculations were performed by using the ASTM D 4015. Generic shear modulus $G/G_0$ using the dynamic triaxial compression method was determined assuming that the initial shear modulus value at low strains $G_0$ equaled the value obtained during low-amplitude oscillations in the resonant column tests. The results of estimating generic shear modulus $G/G_0$ and damping coefficient $D$ in the relationship with shear strains $\gamma$ under the results of triaxial testing were combined with the results of testing under the method of low-amplitude oscillations in the resonant column on the general chart to derive the given dependencies in the range of shear strains $\gamma$ from $10^{-4}$ to $10^{-2}$%. Particular values obtained during each experiment are presented in Appendix A.

### 3. Results and Discussions

The obtained empirical dependencies of generic shear modulus $G/G_0$ and damping coefficient $D$ on shear strains $\gamma$ (Figure 3, Table 2) were obtained during the experiments and approximated by the following logarithmic functions:

$$\frac{G}{G_0} = -A_1 \cdot \ln(\gamma) + B_1 \tag{1}$$

$$D = A_2 \cdot \ln(\gamma) + B_2 \tag{2}$$

where $A_1 = 0.066$, $A_2 = 0.588$, $B_1 = 0.073$, and $B_2 = 11.059$ dimensionless empirical coefficients were assumed with the experimental data.

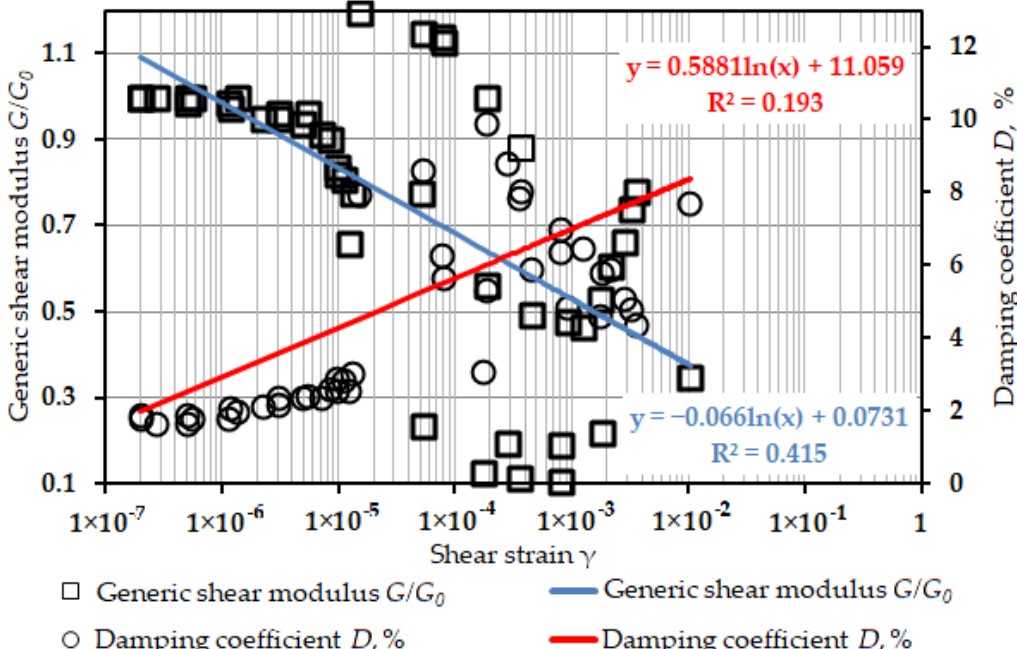

**Figure 3.** Summary graph of dependencies of generic shear modulus $G/G_0$ (unit fractions) and damping coefficient $D$ (%) on the amplitude of shear strains $\gamma$ (unit fractions) for cement-stabilized soils (according to the results of six readings).

**Table 2.** Investigated dynamic properties for the investigated cement-stabilized soils.

| S. No. | Depth, m | $G_0$, MPa | $V_s$, m/s | Relative Shear Strain $\gamma$, Unit Fraction | | | | | | | | | |
|---|---|---|---|---|---|---|---|---|---|---|---|---|---|
| | | | | Shear Modulus $G$, MPa | | | | | | | | | |
| | | | | Damping Coefficient $D$, % | | | | | | | | | |
| 1 | 10.2–10.6 | 240 | 390 | $2.8 \times 10^{-7}$ | $5.0 \times 10^{-7}$ | $1.2 \times 10^{-6}$ | $2.3 \times 10^{-6}$ | $4.9 \times 10^{-6}$ | $8.6 \times 10^{-6}$ | $1.1 \times 10^{-5}$ | $1.5 \times 10^{-5}$ | $8.0 \times 10^{-5}$ | $3.6 \times 10^{-4}$ |
| | | | | 240 | 240 | 237 | 228 | 225 | 217 | 194 | 287 | 270 | 28 |
| | | | | 1.65 | 1.65 | 1.79 | 2.15 | 2.35 | 2.63 | 2.83 | 7.95 | 5.66 | 7.87 |
| 2 | 18.1–18.3 | 263 | 424 | $2.0 \times 10^{-7}$ | $5.6 \times 10^{-7}$ | $1.4 \times 10^{-6}$ | $3.0 \times 10^{-6}$ | $7.3 \times 10^{-6}$ | $9.7 \times 10^{-6}$ | $1.3 \times 10^{-5}$ | $7.9 \times 10^{-5}$ | $1.9 \times 10^{-4}$ | $3.7 \times 10^{-4}$ |
| | | | | 263 | 263 | 263 | 250 | 240 | 216 | 173 | 300 | 263 | 231 |
| | | | | 1.81 | 1.78 | 1.98 | 2.38 | 2.39 | 2.57 | 2.56 | 6.28 | 5.31 | 8.05 |
| 3 | 19.0–19.3 | 244 | 391 | $2.0 \times 10^{-7}$ | $5.1 \times 10^{-7}$ | $1.2 \times 10^{-6}$ | $3.1 \times 10^{-6}$ | $5.6 \times 10^{-6}$ | $9.7 \times 10^{-6}$ | $1.3 \times 10^{-5}$ | $5.3 \times 10^{-5}$ | $1.8 \times 10^{-4}$ | $8.1 \times 10^{-4}$ |
| | | | | 244 | 241 | 237 | 234 | 234 | 204 | 190 | 280 | 32 | 26 |
| | | | | 1.90 | 1.90 | 2.11 | 2.17 | 2.43 | 2.89 | 3.02 | 8.63 | 3.10 | 6.98 |
| 4 | 3.7–4.0 | - | - | $5.4 \times 10^{-5}$ | $2.8 \times 10^{-4}$ | $8.1 \times 10^{-4}$ | $1.8 \times 10^{-3}$ | $1.0 \times 10^{-2}$ | - | - | - | - | - |
| | | | | 58 | 48 | 47 | 54 | 85 | - | - | - | - | - |
| | | | | 17.00 | 8.80 | 6.37 | 5.79 | 7.69 | - | - | - | - | - |
| 5 | 7.1–7.4 | - | - | $1.3 \times 10^{-3}$ | $2.2 \times 10^{-3}$ | $2.8 \times 10^{-3}$ | $3.3 \times 10^{-3}$ | $3.7 \times 10^{-3}$ | - | - | - | - | - |
| | | | | 114 | 148 | 162 | 181 | 191 | - | - | - | - | - |
| | | | | 6.49 | 5.90 | 5.08 | 4.81 | 4.37 | - | - | - | - | - |
| 6 | 9.0–9.4 | - | - | $5.2 \times 10^{-5}$ | $1.9 \times 10^{-4}$ | $4.5 \times 10^{-4}$ | $9.3 \times 10^{-4}$ | $1.8 \times 10^{-3}$ | - | - | - | - | - |
| | | | | 190 | 137 | 121 | 117 | 130 | - | - | - | - | - |
| | | | | 23.42 | 9.93 | 5.90 | 4.87 | 4.60 | - | - | - | - | - |

The selection of a logarithmic function is preconditioned by the maximal acceptability of approximation assessed by the parameter coefficient of determination $R^2$. The necessity of applying similar approximating functions is supported by the further use of laboratory test results for extrapolation on the broader range of shear strains $\gamma$. The problem is that, for the whole range of shear strain programmed by the software, it is not always possible to derive the dependencies of shear modulus degradation and damping coefficient on the shear strain shown in Figure 3. Some gaps in these graphs could be filled utilizing data interpolation obtained in the adjacent lots by employing Functions (1) and (2). This approach is likely to prove its value for further engineering analysis, as, for instance, presented in the work of Siretean et al. (2014) [24].

For the further analysis and description on nonlinear relationships between stresses and strains in soil cement at dynamic loading (to a successful application for numerical analysis of the stress–strain state of modified footings), the resilient-elastic models of Ramberg–Osgood [25] and Hardin–Drnevich (Drnevich et al., 1978) are proposed [22] for the approximation of soil stiffness and damping coefficient. On the basis of the results of laboratory tests, generalized empirical dependencies were drawn that were further approximated by the proposed models. The selected models are widely applicable in nonlinear calculations of the stress–strain state when a footing interacts with a structure while accounting for various dynamic impacts (Solberg et al., 2016) [26,27], (Wolf and Song, 2002) [28], (Ueng and Chen, 1992) [29]. The given models are predominantly used in the model software complexes for the mathematical modelling of the system's footing–structure interaction under static and dynamic loading. However, a conventional method for determining parameters on the basis of experimental data does not exist.

The first proposed model, the Hardin–Drnevich, is a hyperbolic model with three variable parameters: the generic shear modulus $G/G_0$, the relationship between shear strain and nonlinearity limit $\gamma/\gamma_{0.7}$ and empirical parameter $\alpha$ determining the type of nonlinear dependency. Generic shear modulus $G/G_0$ at cyclic loading for this model corresponds to the formula:

$$\frac{G}{G_0} = \frac{1}{1 + \alpha\left(\frac{\gamma}{\gamma_{0.7}}\right)} \tag{3}$$

In the Hardin–Drnevich model, the relationship between generic shear modulus $G/G_0$ and damping coefficient $D$ is described by the following formula:

$$D = \frac{4}{\pi}\left(\frac{1}{1 - \frac{G}{G_0}}\right)\left(1 + \frac{\frac{G}{G_0}}{1 - \frac{G}{G_0}}\ln\left(\frac{G}{G_0}\right)\right) - \frac{2}{\pi} \tag{4}$$

In the current research, in a soil-cement sample from the footing of the designed nuclear power plant, the value of 0.05 was the most optimal for parameter $\alpha$ in the Hardin–Drnevich model. Applying the approximation function of Equations (3) and (4), the curves of the dependency of soil stiffness and soil damping on the shear strain were produced. The results are presented in Figure 4 and Table 3.

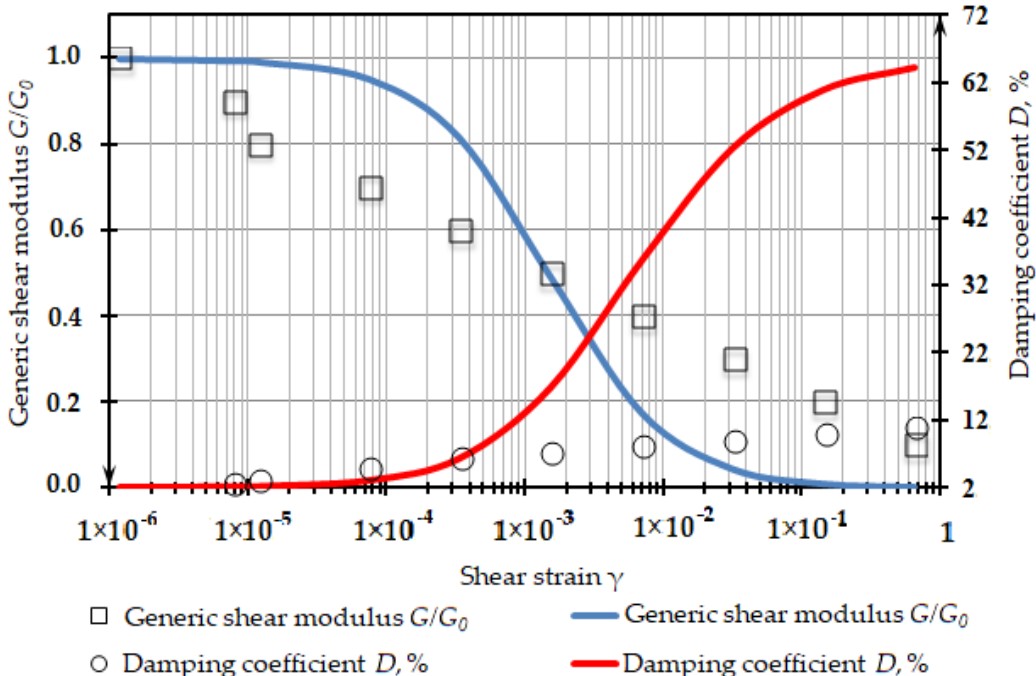

**Figure 4.** Approximated graph of the relationship between generic shear modulus $G/G_0$ (u.f.) and damping coefficient $D$ (%), and shear strain amplitude $\gamma$ (unit fractions) for soil cement with the Hardin–Drnevich model (results of six tests).

The second considered model, namely, the Ramberg–Osgood, includes four parameters: generic shear modulus $G/G_0$, the relationship between the current shear strain and nonlinearity limit $\gamma/\gamma_{0.7}$, and empirical parameters $\alpha$ and $r$, which define the type of nonlinear dependency in the given model. In the Ramberg–Osgood model, the defining dependency corresponds to the formula:

$$\frac{G}{G_0} = \frac{1}{1 + \alpha \left(\frac{\gamma}{\gamma_{0.7}}\right)^{r-1}} \tag{5}$$

where $\gamma$ is shear strain, $\gamma_{0.7}$ is shear strain at which $G/G_0 = 0.722$, $\alpha$ and $r$ are empirical parameters determined from the results of experimental data approximation.

In the Ramberg–Osgood model, the relationship between generic shear $G/G_0$ and damping coefficient $D$ is estimated with the following formula:

$$D = \frac{2}{\pi} \frac{r-1}{r+1} \left(1 - \frac{G}{G_0}\right) \tag{6}$$

The implemented experimental studies show that, for the examined soil cement, the values of parameters $\alpha$ and $r$ for the Ramberg–Osgood model were assumed to be equal to 0.35 and 1.35, respectively. From these parameters, the most optimal approximation of experimental data of laboratory tests is achieved. To obtain these data, it is necessary to combine the results of tests on various laboratory apparatuses, as each has its own limited range of measurement of shear strain measurement in dynamic mode.

**Table 3.** Table of recommended values of dynamic properties.

| N | Soil's Model | Initial Shear Modulus $G_0$, MPa | Shear Modulus $G_{0.722}$, MPa | Shear Strain $\gamma_{0.7}$, Unit Fraction | Velocity $V_s$, m/s | Relative Shear Strain $\gamma$, Unit Fraction / Generic Shear Modulus $G/G_0$, Unit Fraction / Damping Coefficient $D$, % | | | | | | | | | |
|---|---|---|---|---|---|---|---|---|---|---|---|---|---|---|---|
| 1 | Ramberg–Osgood | | | | | $1.2 \times 10^{-6}$ | $8.0 \times 10^{-6}$ | $1.2 \times 10^{-5}$ | $7.5 \times 10^{-5}$ | $3.4 \times 10^{-4}$ | $1.6 \times 10^{-3}$ | $7.1 \times 10^{-3}$ | $3.2 \times 10^{-2}$ | $1.5 \times 10^{-1}$ | $6.7 \times 10^{-1}$ |
| | | | | | | 0.9 | 0.9 | 0.8 | 0.7 | 0.6 | 0.5 | 0.4 | 0.3 | 0.2 | 0.1 |
| | | 244 | 180 | $7.50 \times 10^{-5}$ | 391 | 2.7 | 3.3 | 3.4 | 4.4 | 5.5 | 6.7 | 7.9 | 9.0 | 9.8 | 10 |
| 2 | Hardin–Drnevich | | | | | $1.2 \times 10^{-6}$ | $8.0 \times 10^{-6}$ | $1.2 \times 10^{-5}$ | $7.5 \times 10^{-5}$ | $3.4 \times 10^{-4}$ | $1.6 \times 10^{-3}$ | $7.1 \times 10^{-3}$ | $3.2 \times 10^{-2}$ | $1.5 \times 10^{-1}$ | $6.7 \times 10^{-1}$ |
| | | | | | | 1.0 | 0.9 | 0.9 | 0.9 | 0.8 | 0.5 | 0.2 | 0.1 | 0.1 | 0.1 |
| | | | | | | 2.0 | 2.1 | 2.1 | 3.0 | 6.3 | 16 | 36 | 52 | 60 | 64 |

Using the approximation functions of the Ramberg–Osgood model (5) and (6), the curves of dependency of stiffness and soil intake on shear strain that are recommended for use in succeeding calculations were obtained. The curves are presented in Figure 5 and Table 3.

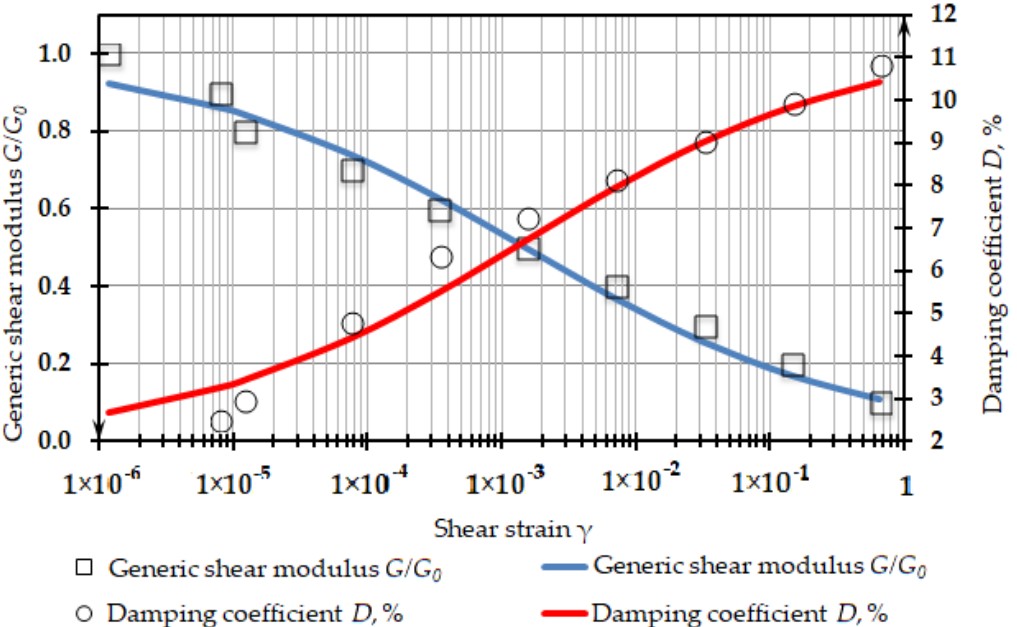

**Figure 5.** Approximated graph of the relationship between generic shear modulus $G/G_0$ (unit fractions) and damping coefficient (%) and shear strain amplitude $\gamma$ (unit fractions) for the soil cement with the Ramberg–Osgood model (results of six tests).

The obtained curves of the dependency of soil stiffness on shear strain had the distinctive S-shape of stiffness reduction (which, on the logarithmic scale of strains, becomes a straight line). This shape is typical for both dispersive soils and specimens modified by the deep soil mixing method. Experimental data analysis showed the minimal strain that can be accurately measured at standard laboratory tests (we assumed the test of triaxial compression or odometer tests as the standard laboratory test). Soil stiffness is often decreased by more than a half its initial value at small strains. Once again, this proves the necessity to analyze the change in the mechanical properties of the soils at various values of shear strain.

The Hardin–Drnevich model is a reasonable fit to the range of ultrasmall strains. However, with a strain increase, this model tends to unreasonably overestimate the damping coefficient. This result is clearly demonstrated in Figure 4, where the curve of the dependency of the damping coefficient sharply went upwards in the range of large strains. This shortcoming of this model is related to its limited flexibility, as it only depends on the constrained number of parameters, which do not allow for the curve's shape to vary in the ample range. This fact does not allow for recommending this model as an analytical model to assess the stress–strain state by the method of mathematical modelling for a soil base modified by the method of deep soil mixing.

At the same time, the Ramberg–Osgood model demonstrated an improved convergence with the results of the experimental studies. The graphs of the degradation of shear strain modulus and damping coefficient on the base of the Ramberg–Osgood model (Figure 5) practically coincided with the points of the experimental readings. The analyzed and obtained curves show that the increase in parameter $r$ in the Ramberg–Osgood model would lead to a worsening of modulus degradation with the increase in shear strain, whereas degradation started at a higher strain value. On the other end of the spectrum, when parameter $\alpha$ was increased, the curve was shifted towards the left side of the diagram, i.e., the deterioration started at lower values of shear strain. Thus, the four-parameter

Ramberg–Osgood model is preferable for use in the numerical modelling of buildings' stress–strain state of modified soil-cement bases.

The investigation of dispersive soils resulted in a decrease in soil stiffness from small to large strains related to the loss of intermolecular and superficial forces inside the soil skeleton. The mechanism of reducing soil-cement stiffness has not been studied thoroughly, as this material is at the nexus of the dynamic behavior theory of soil and concrete. However, when evaluating the behavior of cement-stabilized soils under dynamic loading impacts, the relationships between elastic and plastic deformations could be applied, with retained strength and the typical vibrocreep for both soil and concrete. When the loading direction was reversed, soil and concrete restored their stiffness to the maximal value, approximately equal to the initial soil stiffness. Further, stiffness was reduced again when the loading was imposed in the opposite direction.

## 4. Conclusions

The current research determined the dynamic properties of cement-stabilized soil samples obtained from an industrial building's soil basis. Specific laboratory tests of cement-stabilized soils were carried out using the methods of low-amplitude oscillations in the resonant column and dynamic triaxial compression to study the nonlinear behavior of soils exposed to dynamic loading, particularly the nonlinear change in shear modulus $G$ and damping coefficient $D$ as a function of shear strain $\gamma$. It was required to fix the change of given parameters in the range of shear strains $10^{-4}$–1% with the purpose of further use for the analytical and numerical calculations of interaction of a designed structure with a soil base modified by the DSM method. The results of the research are summarized as follows:

1. The modified soil specimens were tested to their complete saturation with distilled water. The tests were implemented under a consolidated–undrained scheme. The consolidation pressure was specified according to the ground pressure on the sampling depth and consolidation isotropic.

2. As the result of the tests, we obtained the values of velocity of transversal waves $V_s$, initial shear modulus $G_0$; shear strains $\gamma_{0.7}$ (at which $\frac{G}{G_0=0.722}$), dynamic shear modulus $G$, and absorption (damping) coefficient $D$ of soils in the range of shear strains $\gamma$, which where $10^{-4}$ to $10^{-2}$%.

3. The integrated degradation curves of shear modulus and damping coefficient for stabilized soils were obtained by reconciling the test results for soil-cement specimens implemented by two methods—dynamic triaxial compression and low-amplitude oscillations in the resonant column.

4. The Hardin–Drnevich model is a reasonable fit to the range of ultrasmall strains only. With the strain increase, the model tended to unreasonably overestimate the damping coefficient. This fact does not allow for recommending this as a computational model for succeeding numerical modelling. The second considered model of Ramberg–Osgood showed better convergence with the results of experimental research. The curves of degradation of shear modulus and damping coefficient on the base of this model practically coincided with the experimental points. The four-parameter model of Ramberg-Osgood is better for the application in the numerical modelling of the strain–stress state of transformed soil-cement bases of buildings.

5. The empirical linear dependency of $G/G_0$ and $D$ on shear strain $\gamma$ proposed in this study could be further used for the dynamic analysis of structures on a soil base modified by the DSM method for preliminary calculations or by scenarios with data of direct field investigation.

**Author Contributions:** Conceptualization, methodology, investigation, software, A.T.-M. and E.S.; formal analysis, E.S. and A.T.-M.; writing—original draft preparation, E.S. and V.S.; writing—review and editing, all authors; visualization, E.S.; supervision, A.T.-M. and V.S. All authors have read and agreed to the published version of the manuscript.

**Funding:** This work was financially supported by the Ministry of Science and Higher Education (grant 075-15-2021-686). All tests were carried out using research equipment of The Head Regional Shared Research Facilities of the Moscow State University of Civil Engineering.

**Institutional Review Board Statement:** Not applicable.

**Informed Consent Statement:** Not applicable.

**Data Availability Statement:** The data used to support the findings of this study are included within the article. The original details of the data presented in this study are available on request from the corresponding author.

**Conflicts of Interest:** The authors declare no conflict of interest.

**Appendix A**

Laboratory tests of reinforced soils were performed using low-amplitude oscillations in a resonant column and dynamic triaxial compression to study the nonlinear behavior of soils under dynamic loads, particularly the nonlinear change in shear modulus $G$ and damping coefficient $D$ depending on shear deformations $\gamma$.

Laboratory tests of soil-cement samples in the mode of low-amplitude dynamic tests in a resonant column (triaxial static loading with simultaneous excitation of low-amplitude torsional vibrations in the sample) were carried out in accordance with the requirements of Russian State Standard GOST 12248-2010 "Soils. Methods for laboratory determination of strength and deformability characteristics" (https://docs.cntd.ru/document/1200084869; accessed on 1 June 2022) and Russian State Standard GOST 56353-2015 "Soils. Methods for laboratory determination of the dynamic properties of dispersed soils" (https://docs.cntd.ru/document/1200118271; accessed on 1 June 2022). Samples of reinforced soils were tested at full water saturation with distilled water. The tests were carried out according to the consolidated–undrained scheme. The consolidation pressure was set in accordance with the natural pressure at the sampling depth. Consolidation took place in isotropic mode.

Laboratory tests of soil cement samples in the mode of dynamic triaxial compression (triaxial static loading with simultaneous excitation of vertical vibrations in the sample) were carried out in accordance with the requirements of Russian State Standard GOST 12248-2010 "Soils. Methods for laboratory determination of strength and deformability characteristics" and Russian State Standard GOST 56353-2015 "Soils. Methods for laboratory determination of the dynamic properties of dispersed soils". Samples of reinforced soils were tested at full water saturation with distilled water. The tests were carried out according to the consolidated–undrained scheme. Consolidation pressure was set in accordance with the natural pressure at the sampling depth. Consolidation took place in isotropic mode.

As a result of the tests, the values of shear wave velocity Vs (m/s), initial shear modulus $G_0$ (MPa), shear strain $\gamma_{0.7}$ (fraction units), dynamic shear modulus $G$ (MPa), and absorption (damping) coefficient $D$ (%) soils were in the range of shear deformations $\gamma$ $10^{-4}$–1 %. Particular values of mechanical (dynamic) properties for each tested soil sample are contained in this appendix in Figures A1–A6.

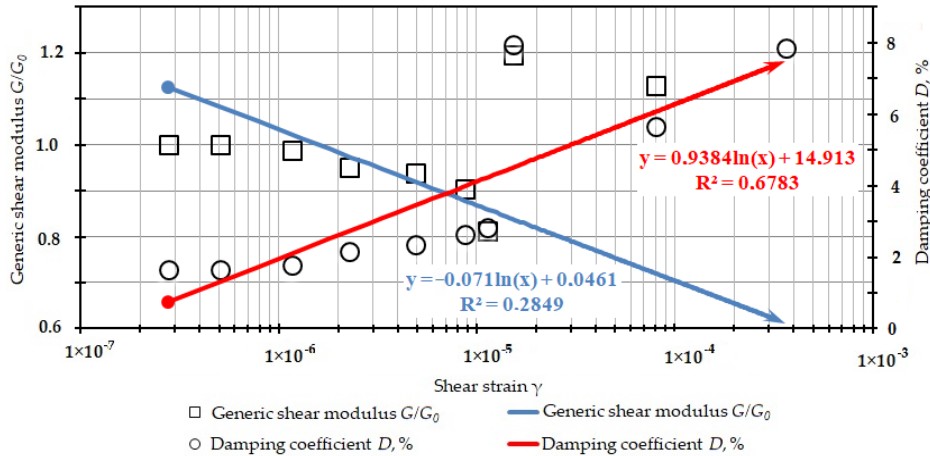

**Figure A1.** Graph of dependencies of generic shear modulus $G/G_0$ (unit fractions) and damping coefficient $D$ (%) on the amplitude of shear strains $\gamma$ (unit fractions) for cement stabilized soils. Test 1 (resonant column).

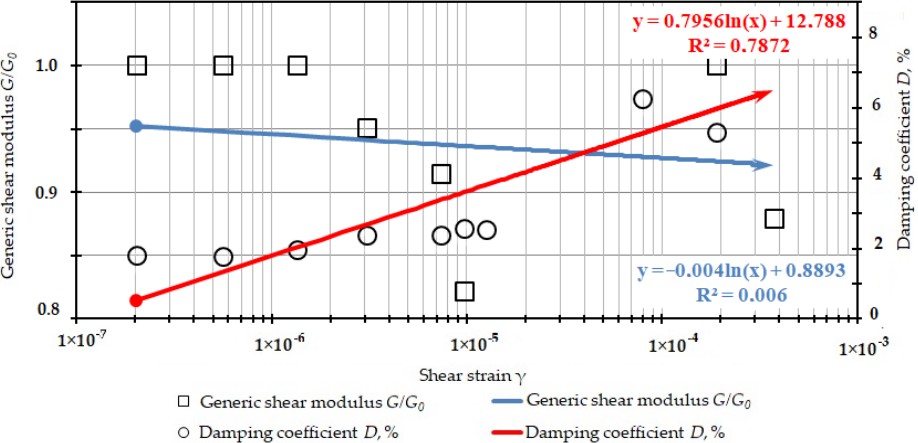

**Figure A2.** Graph of dependencies of generic shear modulus $G/G_0$ (unit fractions) and damping coefficient $D$ (%) on the amplitude of shear strains $\gamma$ (unit fractions) for cement stabilized soils. Test 2 (resonant column).

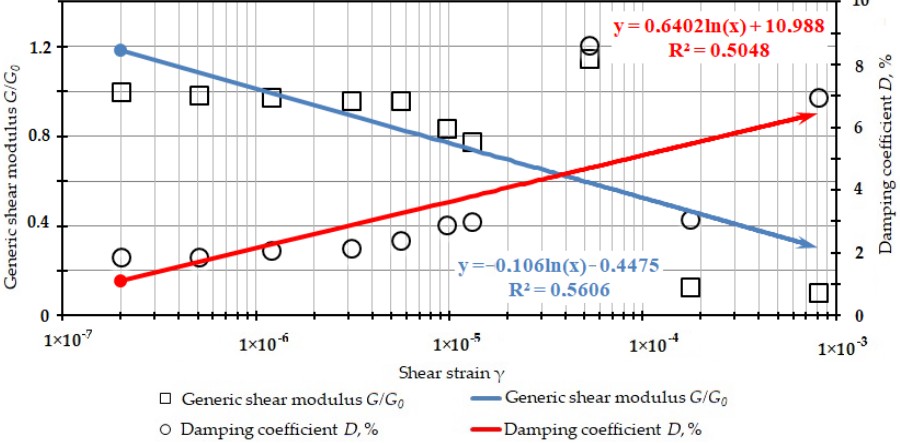

**Figure A3.** Graph of dependencies of generic shear modulus $G/G_0$ (unit fractions) and damping coefficient $D$ (%) on the amplitude of shear strains $\gamma$ (unit fractions) for cement stabilized soils. Test 3 (resonant column).

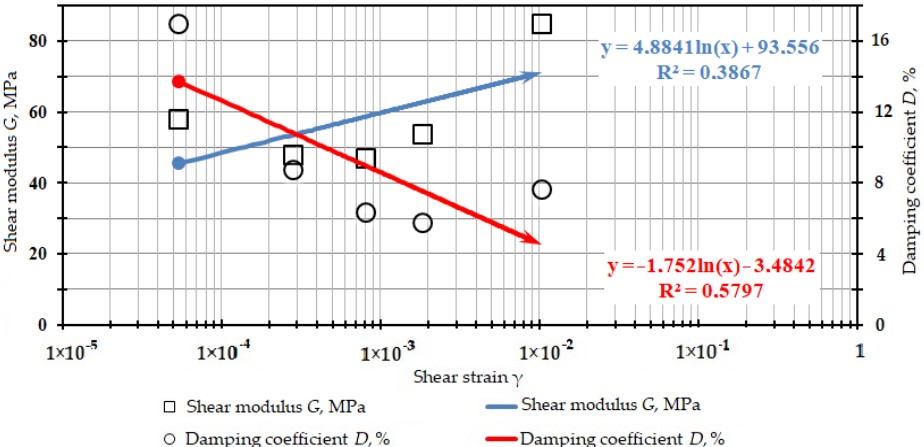

**Figure A4.** Graph of dependencies of generic shear modulus $G/G_0$ (unit fractions) and damping coefficient $D$ (%) on the amplitude of shear strains $\gamma$ (unit fractions) for cement stabilized soils. Test 4 (triaxial dynamic compression).

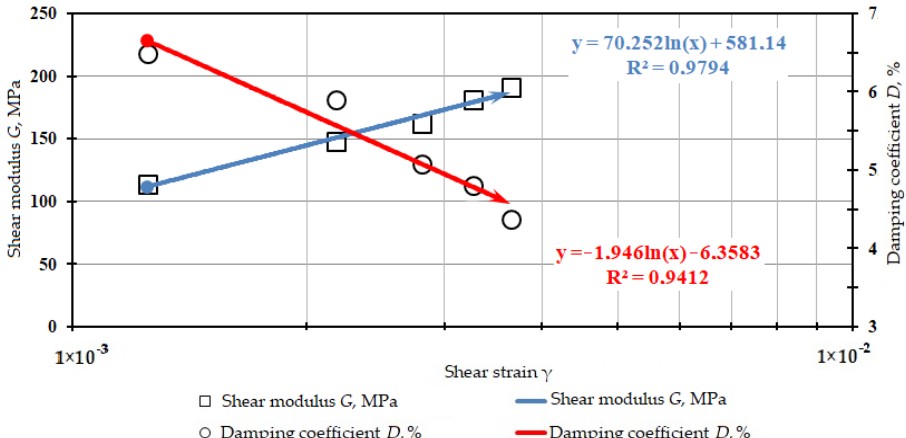

**Figure A5.** Graph of dependencies of generic shear modulus $G/G_0$ (unit fractions) and damping coefficient $D$ (%) on the amplitude of shear strains $\gamma$ (unit fractions) for cement stabilized soils. Test 5 (triaxial dynamic compression).

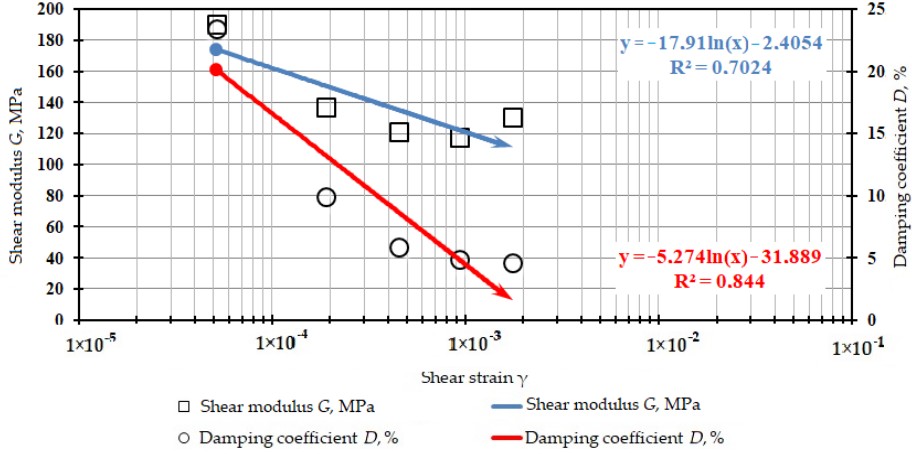

**Figure A6.** Graph of dependencies of generic shear modulus $G/G_0$ (unit fractions) and damping coefficient $D$ (%) on the amplitude of shear strains $\gamma$ (unit fractions) for cement stabilized soils. Test 6 (triaxial dynamic compression).

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
