# Peer review of "Dynamic Properties of Soil Cements for Numerical Modelling of the Foundation’s Basis Transformed under the Technology of Deep Soil Mixing: A Determination Method"

_buildings, doi:10.3390/buildings12071028_

Round 1

Reviewer 1 Report

Please see comments.

Author Response

The author's team thanks for the detailed consideration of the work and the proposed recommendations for the article. The article with the changes and additions made has been prepared for re-acquaintance.

Reviewer 2 Report

This paper investigated the dynamic  properties of soil-cements, and concluded the four-parameter model of Ramberg-Osgood is preferred in numerical modelling. The work is of great significance and the paper is well organized. However, I suggest the authors to refine the background description in the Introduction part. 

Author Response

(The authors gave the same response as above.)

Reviewer 3 Report

 The article provides a laboratory-scale investigation of the dynamic mechanical properties of soil-cement stabilized specimens ranging from ultra-small to large values of shear strain.

 1-     The language of the article is understandable. The gained results and proposed methods are of high interest to readers in geotechnical fields as it provides an approximation and reference calculation methods of the dynamic properties of the soil stabilized by cement for both scientific research and practical applications.

 2-     The title of the article should be corrected as the following as it will be more attractive to the readers in the geotechnical field, and it more coincides with the main aim of the conducted research: "Dynamic properties of soil-cements for numerical modeling of foundations transformed under the technologies of Deep Soil Mixing: A determination method."

 3-     Standards and regulatory documents should be described so that a foreign language reader can understand them (for example, Russian State Standard GOST 10180-2012 "Concretes. Methods for strength determination using reference specimens"  https://docs.cntd.ru/document/1200100908 ).

4-     In the future, we highly recommend that the authors generalize the gained results to validate them to be used for approximation, evaluation, and verification of the actual dynamic properties in field applications under a real scale load applied to the foundations.

 Overall: The gained results, the materials and methods used for conducting the research, and the experimental results gained are valuable and sufficient, allowing us to recommend this article for publications after considering the given minor remarks.

Author Response

(The authors gave the same response as above.)

Round 2

Reviewer 1 Report

English language and style are fine/minor spell check required